# Multimodal Social Interaction with Multi-speaker Attention Alignment

## Abstract

Understanding social interaction in video requires reasoning over a dynamic interplay of verbal and non-verbal cues: who is speaking, to whom, and with what gaze or gestures. While Multimodal Large Language Models (MLLMs) are natural candidates, simply adding visual inputs yields surprisingly inconsistent gains on social tasks. Our quantitative analysis of cross-modal attention inside state-of-the-art MLLMs reveals a core failure mode: in multi-speaker scenes, visual and textual tokens lack speaker-consistent alignment, exhibiting substantially weaker cross-modal attention than in object-centric images. To address this, we propose a multimodal multi-speaker attention alignment method that can be integrated into existing MLLMs. First, we introduce *dynamic cross-modal head selection* to identify attention heads most responsible for grounding. Then, an *adaptive social-aware attention bias*, computed from existing attention patterns and speaker locations, is injected into the attention mechanism. This bias reinforces alignment between a speaker's visual representation and their utterances without introducing trainable parameters or architectural changes. Experiments on three datasets (TVQA+, MMSI, and OnlineMMSI) across four social tasks demonstrate that our approach improves the ability of MLLMs and achieves state-of-the-art results on multiple tasks. Attention visualizations confirm our method successfully focuses the model on speaker-relevant regions, enabling more robust multi-party social reasoning.

## 1 Introduction

Understanding social interaction requires modeling multi-party human behaviors through both verbal and non-verbal cues, including dialogue, gestures (Cao et al., 2025), gaze (Zhou et al., 2024), and facial expressions (Hyun et al., 2024). To study these interactions, prior works have proposed a variety of tasks and benchmarks, such as video question answering (VQA), speaking target detection, mentioned player prediction, and pronoun coreference resolution (Lei et al., 2020; Lee et al., 2024a). Beyond serving as evaluation platforms, these tasks underpin socially intelligent AI agents that operate in real-world multi-party scenarios like board games, daily conversations, and meetings.

Given their ability to comprehend both verbal and non-verbal information, multimodal large language models (MLLMs) are natural candidates for these tasks (Lee et al., 2024a; Li et al., 2025a; Park et al., 2025b). However, our analysis reveals a critical limitation: the addition of visual information does not consistently improve, and can even degrade their performance in multi-person settings. For example, on OnlineMMSI (Li et al., 2025a), supplying video frames to Qwen2.5-VL (Bai et al., 2025) input yields no gain on the mentioned player prediction task, while LLaMA-3.2-Vision (Dubey et al., 2024) sees its performance drop on the pronoun coreference resolution task (Li et al., 2025a). These observations suggest that current MLLMs struggle to effectively exploit multimodal cues in complex multi-person social settings.

To better understand why MLLMs fail to leverage multimodal cues, we conduct a systematic quantitative analysis of cross-modal attention weights inside state-of-the-art MLLMs (Bai et al., 2025). By measuring the attention weights between a speaker's textual tokens and their corresponding visual region (i.e., their bounding box), we uncover a stark deficiency. We find that the cross-modal alignment in multi-person videos is significantly weaker and less focused compared to the alignment observed in general object-centric datasets like COCO (Lin et al., 2014). This limitation results in

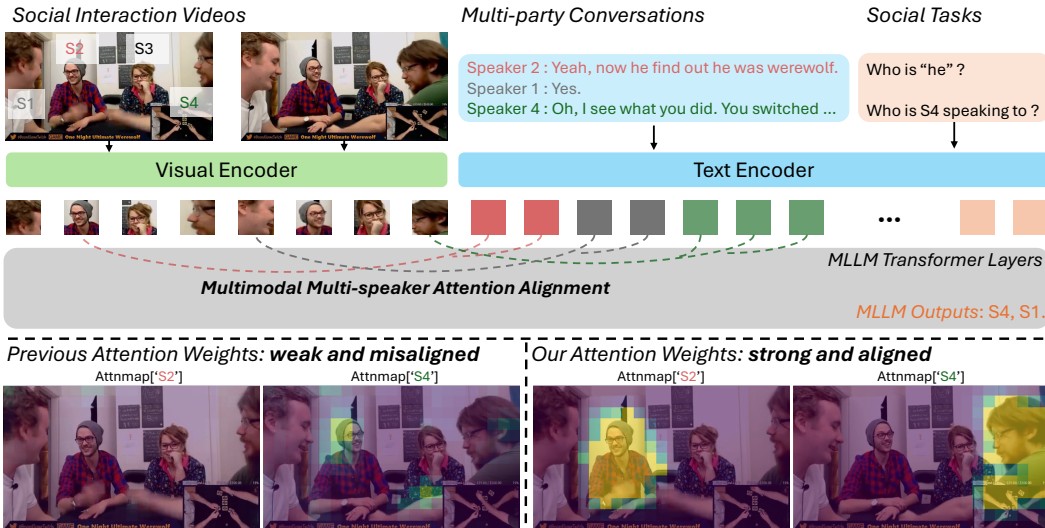

Figure 1: We propose a multimodal multi-speaker attention alignment method for MLLMs to understand social interactions in videos. Visualization of cross-attention weights in transformer layers confirms that our approach strengthens the model's focus on areas relevant to the active speaker.

inconsistent alignment between visual and textual modalities, thereby constraining the effectiveness of MLLMs in multi-person social tasks.

To address this misalignment problem, we propose a multimodal multi-speaker attention alignment method. Our approach intervenes directly within the transformer's cross-attention layers. We first propose a **dynamic cross-modal head selection** strategy that identifies attention heads most responsible for visual-text grounding. We then apply an **adaptive social-aware attention bias** to these heads, which amplifies the attention scores between the visual and textual tokens belonging to the same speaker. As illustrated in fig. 1, this mechanism explicitly guides the model to associate the correct visual features with the corresponding dialogue. Crucially, our method requires no additional trainable parameters or architectural changes in models.

We evaluate our method on three multimodal social interaction benchmarks (TVQA+ (Lei et al., 2020), MMSI (Lee et al., 2024a), and OnlineMMSI (Li et al., 2025a)) across four representative tasks. Integrated into Qwen2.5-VL (Bai et al., 2025), our method consistently outperforms strong baselines, yielding an average accuracy improvement of 3.5%. It achieves state-of-the-art performance on three task settings and remains highly competitive on the remaining one. Attention visualizations further confirm that our approach successfully guides the model to focus on speaker-relevant regions in videos.

Our main contributions are summarized as follows:

- We are the first to systematically quantify and identify the cross-modal attention misalignment in MLLMs as a key bottleneck for understanding multi-party social interactions.
- We propose a novel attention alignment method that dynamically reinforces the association between speakers' visual and textual representations without additional trainable parameters.
- Extensive experiments demonstrate that our method effectively guides model attention to speaker-relevant regions, thereby improving performance in diverse multimodal social interaction tasks.

## 2 RELATED WORKS

### 2.1 MULTIMODAL SOCIAL INTERACTION

Multimodal social interaction refers to human communication across multiple modalities, including spoken language, facial expressions (Hyun et al., 2024), gaze (Zhou et al., 2024), gestures (Cao et al., 2025), and body movements (Balazia et al., 2022). Prior research has proposed a variety of

related tasks and benchmarks, such as video question answering (VQA) (Lei et al., 2018; Zadeh et al., 2019; Hyun et al., 2024; Mathur et al., 2025; Kong et al., 2025), conversational modeling (Ryan et al., 2023; Lee et al., 2024a; Jia et al., 2024; Chang et al., 2025), speaker prediction (Northcutt et al., 2020; Müller et al., 2021), and social behavior classification (Lai et al., 2023; Cao et al., 2025). These tasks hold strong potential for enabling AI agents to operate in multi-party social scenarios, including board games (Lai et al., 2023; Grauman et al., 2022), daily conversations (Northcutt et al., 2020), and multi-person meetings (Müller et al., 2018; Kraaij et al., 2005). Leveraging MLLMs for such social interaction tasks has recently become an emerging trend (Lee et al., 2024b; Mathur et al., 2024; Mou et al., 2024). This work is the first to introduce a multimodal attention alignment method for multi-person conversations, evaluated across three datasets and four social interaction tasks, showing its capacity to generalize across diverse multimodal social interaction tasks and benchmarks.

## 2.2 MULTIMODAL BIAS AND ALIGNMENT IN MLLMs

In multimodal learning, diverse modalities have been incorporated into MLLMs (Liu et al., 2023; Yan et al., 2024), where one fundamental challenge is achieving effective cross-modal alignment (Radford et al., 2021; Girdhar et al., 2023; Chen et al., 2024c; Amirloo et al., 2024; Li et al., 2025b). Recent studies (Wu et al., 2024b; Amirloo et al., 2024; Xiao et al., 2024; Zheng et al., 2025; Park et al., 2025b; Zhang et al., 2025d) have highlighted that MLLMs are deeply affected by modality bias, where the models' understanding and reasoning capabilities rely heavily on the textual modality while underutilizing other modalities. To mitigate this bias and align modalities, some approaches have focused on collecting additional datasets (Chen et al., 2024a; Wu et al., 2024c; Yue et al., 2024; Chen et al., 2024b), reinforcement learning (Pi et al., 2024; Sun et al., 2024; Zhang et al., 2025b;c), while other methods have sought to adjust the model's attention toward non-text modalities (Xing et al., 2024; Zhang et al., 2024; Tong et al., 2024; Song et al., 2025; An et al., 2025; Tang et al., 2025; Wang et al., 2025; Zhang et al., 2025a). These methods have demonstrated effectiveness on tasks such as VQA, but they lack evaluation and exploration in multi-speaker social interaction scenarios.

Existing work on multimodal social interaction has proposed several strategies for aligning visual and textual modalities across multiple speakers. (Lee et al., 2024a) uses speaker embeddings (Devlin et al., 2019), (Li et al., 2025a) leverages visual prompts (Shtedritski et al., 2023), (Park et al., 2025a) introduces Chain-of-Thought, and (Agrawal et al., 2024) incorporates the audio modality for alignment. Compared to these works on social interactions, our study is the first to systematically and quantitatively investigate this misalignment in social benchmarks. We are also the first to utilize the cross-attention map within transformer layers for multi-person social interaction tasks.

## 3 ANALYSIS OF CROSS-MODAL ALIGNMENT IN MULTI-SPEAKER SETTINGS

Alignment between modalities is a fundamental challenge in vision-language models (VLMs) and multimodal large language models (MLLMs), and a large body of work has focused on learning aligned representations between visual and textual encoders (Radford et al., 2021). This alignment can be quantitatively assessed via the cross-modal attention weights between textual and visual features (Alayrac et al., 2022). When the visual tokens $\mathcal{V}$ and textual tokens $\mathcal{U}$ are concatenated and processed by a transformer, the self-attention mechanism (Vaswani et al., 2017) enables interactions across modalities. Formally, let $\mathcal{X} = [\mathcal{V}; \mathcal{U}] \in \mathbb{R}^{(THW+K) \times d}$ denote the concatenated token sequence. The attention weights are computed as

$$\text{Attn}(i, j) = \text{softmax}_j \left( \frac{(x_i W_Q)(x_j W_K)^\top}{\sqrt{d}} \right), \tag{1}$$

where $x_i, x_j \in \mathcal{X}$ are token embeddings and $W_Q, W_K$ are projection matrices. In the cross-modal case, we specifically focus on the sub-matrix of $\text{Attn}(i, j)$ where $i$ indexes text tokens and $j$ indexes visual tokens. This sub-matrix, denoted as the **cross-modal attention weights**, captures the semantic grounding between textual and visual modalities. High attention weights in this matrix indicate that tokens from text effectively attend to semantically corresponding visual tokens. For example, as illustrated in fig. 2 (a), tokens representing "cat", "car", and "flower" attend strongly to visual tokens corresponding to object regions. Such interpretable cross-modal attention maps have been widely utilized in multimodal tasks, including MLLMs for visual grounding (Wu et al., 2024a; Zhang et al., 2025a) and text-to-image generation models (Chefer et al., 2023; Hertz et al., 2023).

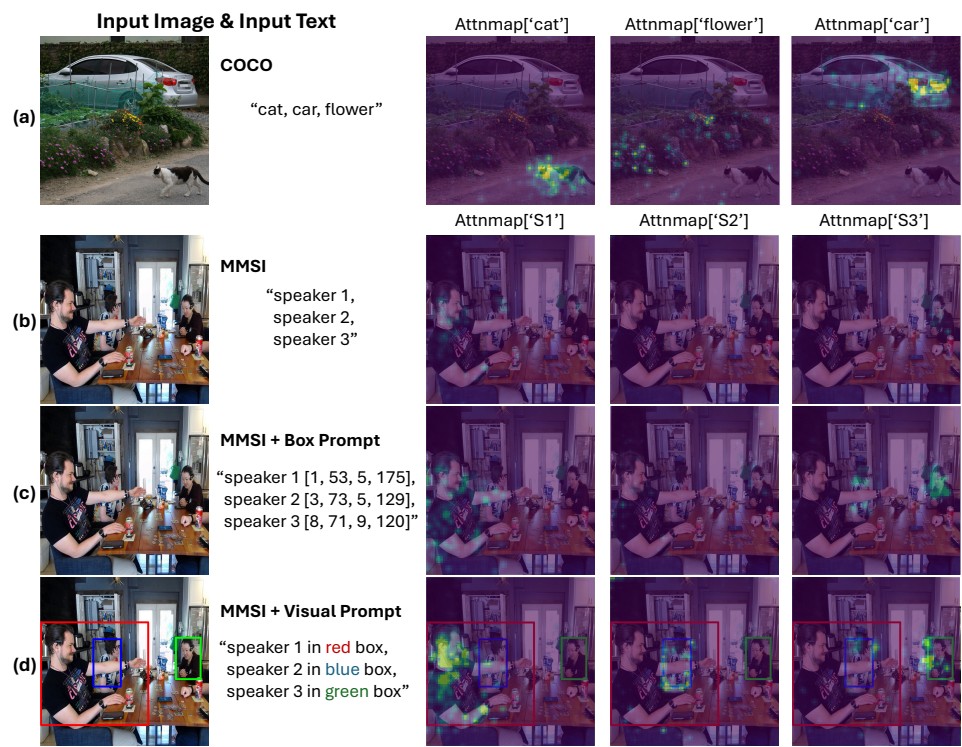

Figure 2: Cross attention weights in Qwen2.5-VL layer 16. Compared to general images, cross-modal alignment in multi-speaker images is weak and inconsistent. Image resolution is 2000x2000.

In multi-speaker social interaction scenarios, challenges arise due to the presence of multiple individuals in the visual scene and ambiguous textual references in conversations. For example, speakers are often mentioned by names or anonymized labels such as "speaker 2", which do not clearly correspond to visual regions. As illustrated in fig. 2 (b), the attention weights of speakers' textual tokens are highly scattered, preventing the model from effectively leveraging the corresponding visual information. One attempt to mitigate this issue is shown in fig. 2 (c), where bounding box coordinates are prompted into the text input. However, we observe that the resulting cross-modal attention remains weak, and the model still struggles to establish clear correspondences. Previous works (Li et al., 2025a; Shtedritski et al., 2023) have also proposed introducing visual prompts, such as adding highlighted bounding boxes or keypoints in the image (fig. 2 (d)). This strategy indeed helps speakers' textual tokens attend to the correct region, but the attention tends to concentrate along the bounding box boundaries rather than the interior. Moreover, we find that the attention map of speaker 3 becomes misaligned, incorrectly overlapping with the region of speaker 2.

To investigate how well MLLMs align textual references with visual evidence in multi-speaker images, we quantitatively analyze cross-modal attention through controlled experiments with Qwen2.5-VL (Bai et al., 2025). Specifically, given a text token $u_i \in \mathcal{U}$ and its corresponding visual tokens $\mathcal{V}_s \subset \mathcal{V}$, we define the alignment score as

$$AttnMax(u_i, \mathcal{V}_s) = \max_{v \in \mathcal{V}_s} \text{Attn}(u_i, v), \quad AttnMean(u_i, \mathcal{V}_s) = \frac{1}{|\mathcal{V}_s|} \sum_{v \in \mathcal{V}_s} \text{Attn}(u_i, v) \quad (2)$$

We compute such statistics across different datasets and compare under various alignment strategies.

**COCO** (Lin et al., 2014). We sample 1,110 images from the COCO object detection validation set, and compute attention with text queries such as "{class 1}, {class 2}, . . .".

**MMSI** (Lee et al., 2024a). We use 1,921 images with queries "{speaker 1}, {speaker 2}, . . .".

**MMSI + Box Prompt** (Bai et al., 2025). The text input is augmented with bounding box coordinates, e.g., "{speaker 1} in [x,y,z,t], {speaker 2} in [a,b,c,d], . . .".

**MMSI + Visual Prompt** (Li et al., 2025a). Bounding boxes are drawn in distinct colors on the image, and the query takes the form "{speaker 1} in red box, {speaker 2} in blue box, . . .".

**Ours**. We apply our proposed multi-speaker alignment method, which explicitly enhances attention weights in speaker-specific regions. See section 4 for details.

Table 1: Cross attention weights in COCO and MMSI images.

| Image Source | Alignment Method | $AttnMax_{\times 10^{-2}}$ | $AttnMean_{\times 10^{-4}}$ |
|---|---|---|---|
| COCO | / | 9.23 | 15.56 |
| MMSI | / | 4.54 | 3.26 |
| | box prompt | 4.49 | 3.93 |
| | visual prompt | 6.29 | 5.29 |
| | Ours | **17.09** | **26.20** |

We report the quantitative results in table 1. Compared to general objects in COCO detection dataset, the attention between images and speaker tokens in MMSI is substantially lower, highlighting the difficulty of aligning speaker references in multi-person contexts. We further observe that introducing visual prompts indeed improves attention weights, but the gains remain limited. This reveals a fundamental challenge for MLLMs: cross-modal alignment for multi-speaker scenarios is weak and inconsistent, as the model struggles to establish clear correspondences between textual references to speakers and their visual representations.

## 4 PROPOSED METHOD

To address the problem of weak and inconsistent cross-modal alignment in social tasks, we propose a multimodal multi-speaker attention alignment method. Our approach consists of two key components: (1) a dynamic cross-modal head selection mechanism that identifies attention heads most relevant for multimodal grounding, and (2) an adaptive social-aware attention bias that reinforces cross-modal token alignment. An overview of the method is illustrated in fig. 3.

**Input for MLLMs.** Let the social interaction video be mapped into a set of visual tokens $\mathcal{V} = \{v_{t,h,w} \in \mathbb{R}^d \mid t \in [1, T], h \in [1, H], w \in [1, W]\}$ by the patch embedder and visual encoder, where each token corresponds to a spatio-temporal patch indexed by $(t, h, w)$. The transcripts consist of speakers' utterances, which are tokenized and encoded into $\mathcal{U} = \{u_k \in \mathbb{R}^d \mid k \in [1, K]\}$, where each token $u_k$ is associated with a speaker label $s$ and a timestamp $t$. In general, the speaker label $s$ is determined by who speaks the utterance, except for certain special tokens that explicitly refer to speakers (e.g., "Mitchell" or "speaker 2"), which are consistently assigned the label of the person they denote. Note that textual contents unrelated to speaker utterances, such as the system prompt and task instructions, are not included in $\mathcal{U}$. In addition, the dataset provides a set of speaker bounding boxes $\mathcal{B} = \{b_{s,t}\}$, where each box $b_{s,t}$ specifies the spatial location of speaker $s$ at frame $t$. By mapping box coordinates to the grid of visual tokens, we obtain subset $\mathcal{V}_{s,t}$ associated with each speaker label.

### 4.1 DYNAMIC CROSS-MODAL HEAD SELECTION

Modern MLLMs employ multi-head attention, with different heads capturing complementary facets of token interactions (Vaswani et al., 2017; Voita et al., 2019). Previous studies (Bi et al., 2025) in MLLMs have identified that specific transformer layers contain specialized "visual heads" that reliably focus on image tokens during task-solving. The presence and focus of such heads vary across models and training strategies, indicating that visual heads are dynamic rather than fixed.

To preserve the pretrained capabilities of MLLMs while improving their cross-modal grounding, we propose a dynamic cross-modal head selection mechanism that identifies the subset of heads with strong cross-modal interactions. Concretely, let $\mathcal{V}_{all} = \bigcup_{s \in S} \bigcup_{t \in T} \mathcal{V}_{s,t}$ denote the set of visual tokens inside bounding boxes for all speakers in the video. We define a threshold $\lambda$ to classify each attention head, based on the cross-modal attention sub-matrix $\text{Attn}(\mathcal{U}, \mathcal{V}_{all})$ that represents the attention from utterance tokens to all speaker regions:

$$head \ is \begin{cases} active, & \frac{1}{|\mathcal{U}|\,|\mathcal{V}_{all}|} \sum_{u \in \mathcal{U}} \sum_{v \in \mathcal{V}_{all}} \text{Attn}_{head}(u, v) > \lambda, \\ inactive, & \text{otherwise.} \end{cases} \tag{3}$$

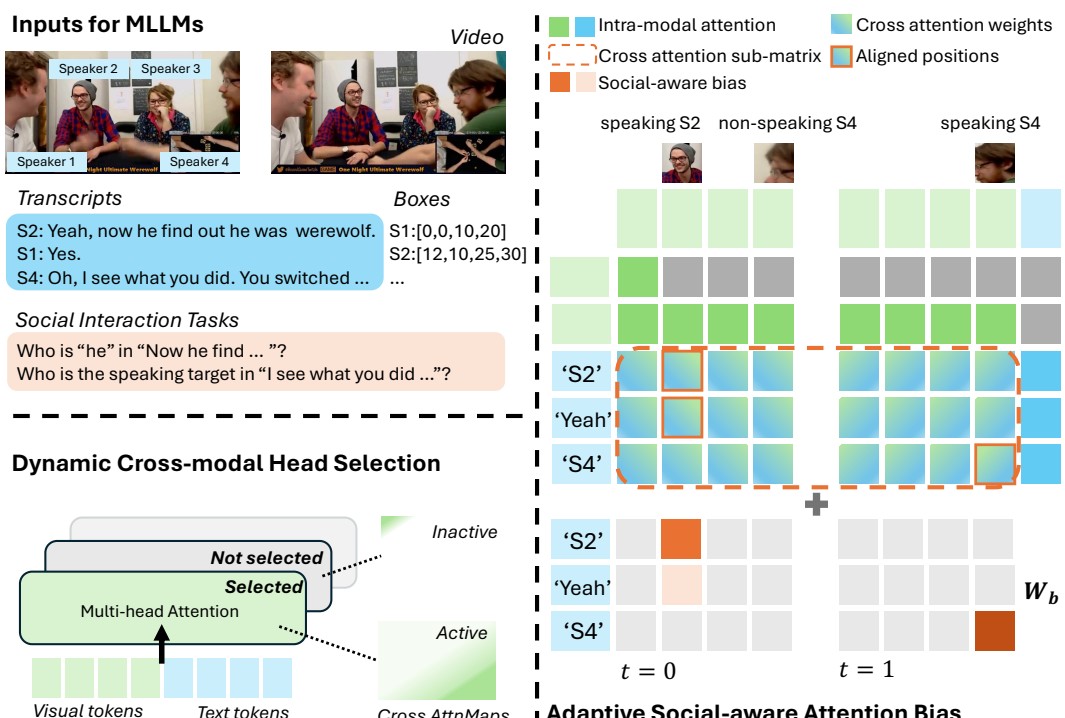

Figure 3: Overview of proposed method.

As illustrated in fig. 3, an *active* head is characterized by having distinctly high attention weights concentrated in one or more speaker regions, whereas an *inactive* head exhibits weak cross-modal attention across all regions. Only active heads are selected for applying the subsequent social-aware attention bias.

## 4.2 ADAPTIVE SOCIAL-AWARE ATTENTION BIAS

In attention computation, adding a bias term to attention weights is a common strategy to control token interactions. For example, language models introduce padding masks or causal masks to prevent tokens from attending to irrelevant or future positions (Devlin et al., 2019; Radford et al., 2019). In the context of social interaction, to strengthen the attention between visual and textual tokens belonging to the same speaker $s$ in frame $t$, we introduce a *social-aware bias* $W_b$ applied within the active heads. Specifically, for a text token $u_i$ associated with speaker $s$, we assign the bias value for each visual token $v_j$ as

$$W_b(u_i, v_j) = \alpha \cdot \max_{v \in \mathcal{V}_{all}} \frac{(u_i W_Q)(v W_K)^\top}{\sqrt{d}}, \quad u_i \in \mathcal{U}_{s,t}, \quad v_j \in \mathcal{V}_{s,t}, \quad (4)$$

where $\alpha$ is a scaling factor controlling the bias strength, and $\max_{v \in \mathcal{V}_{all}} \text{Attn}(u_i, v)$ captures the strongest cross-modal interaction that $u_i$ originally attends to among all speakers' visual tokens.

The motivation of using adaptive weights for different tokens is that certain tokens (e.g., "speaker", "Sheldon", or object mentions) naturally exhibit stronger semantic interactions with visual content, while others (e.g., discourse fillers such as "yeah", "then") are much weaker. By assigning the maximum attention value to speaker-associated regions, we softly shift the distribution of attention towards the visual area of the current speaker, without suppressing the token's original attention pattern. This design ensures that attention alignment is enhanced in a smooth and adaptive way rather than enforced rigidly. Finally, the adjusted attention is computed as:

$$\widetilde{\text{Attn}}(i, j) = \text{softmax}_j\left(\frac{(u_i W_Q)(v_j W_K)^\top}{\sqrt{d}} + W_b(u_i, v_j)\right). \quad (5)$$

Our method requires no additional trainable parameters. Moreover, by leveraging dynamic head selection, it introduces only minimal computational overhead while effectively utilizing speaker bounding box annotations to enhance cross-modal alignment in multi-speaker videos.

## 5 EXPERIMENTS

### 5.1 DATASETS

We conduct experiments on three publicly available datasets under four social task settings. These datasets contain videos, timestamped transcripts, and speaker bounding box annotations, which are utilized in both training and evaluation. The datasets statistics are described below:

**TVQA+** (Lei et al., 2020; 2018) is a multi-party video question answering dataset with rich dynamics and realistic social interactions built on TV series. The QA-pairs are diverse, covering dialogue understanding, reasoning, and speaker relations modeling. In our experiments, we select samples containing at least one annotated speaker bounding box, resulting in 17,306 training samples and 2,211 test samples. On average, each sample involves 1.9 speakers, 23.8 words and 7.8 seconds.

**MMSI** (Lee et al., 2024a) is a recent social interaction benchmark built from multi-party board game videos (Lai et al., 2023) collected from YouTube and Ego4D (Grauman et al., 2022). It defines three challenging tasks to capture fine-grained interaction dynamics: speaking target identification, pronoun coreference resolution, and mentioned player prediction. Following their split and preprocessing, we use the YouTube subset, which contains 7,111 training samples and 1,921 test samples. On average, each sample involves 4.1 speakers, 85.2 words, and 3.0 seconds of video.

**OnlineMMSI** (Li et al., 2025a) is an extension of MMSI that reformulates three tasks under an online setting, where only preceding context of a conversation is available, without access to future dialogue. This design increases task difficulty and enhances practical applicability. The data split and statistics is identical to MMSI, with a forward-shifted historical window applied to each sample.

### 5.2 IMPLEMENTATION DETAILS

We adopt Qwen2.5-VL-Instruct-7B (Bai et al., 2025) as the base MLLM in all experiments. Following dataset annotations (Lei et al., 2020; Lee et al., 2024a), videos are processed at resolution of 640×360 and uniformly sampled into 8 frames. During training, we fine-tune the model using LoRA (Hu et al., 2022) applied to all projection layers. Following (Li et al., 2025a), we set the LoRA rank to 512, the learning rate to 1e-4, the batch size to 4, and train for 5 epochs. All experiments are conducted on a single NVIDIA A100 GPU, with the implementation built on LLaMA-Factory (Zheng et al., 2024) and pytorch (Paszke et al., 2019). We set $\lambda = 5e - 5$ and $\alpha = 1.0$ in our method. The prompts used for MLLM instructions are provided in appendix A.1.

### 5.3 RESULTS

Table 2: Accuracy on TVQA+, MMSI and OnlineMMSI. *T* for speaking target identification, *P* for pronoun coreference resolution, *M* for mentioned speaker prediction. * TLNet/ST-VLM results are taken from their paper, which may adopt a different split from ours. More descriptions of the baselines are provided in appendix A.2.

| Method | TVQA+ | MMSI | | | OnlineMMSI | | |
|---|---|---|---|---|---|---|---|
| | *VideoQA* | *T* | *P* | *M* | *T* | *P* | *M* |
| Random | 20.0 | 21.0 | 23.2 | 23.7 | 21.0 | 23.2 | 23.7 |
| ST-VLM-7B* (Ko et al., 2025) | 68.1 | | | | | | |
| TLNet* (Liang et al., 2024a) | 75.5 | | | | | | |
| MMSI (Lee et al., 2024a) | | **74.5** | 73.0 | 62.5 | 59.1 | 63.4 | 47.3 |
| OnlineMMSI (Li et al., 2025a) | 86.1 | 66.5 | 76.2 | 63.5 | **64.8** | 72.9 | 49.4 |
| Qwen2.5-Text (Bai et al., 2025) | 78.0 | 66.3 | 77.0 | 61.7 | 59.3 | 74.4 | 49.0 |
| Qwen2.5-VL (Bai et al., 2025) | 85.1 | 63.3 | 77.2 | 58.3 | 59.6 | 75.1 | 50.2 |
| **Qwen2.5-VL+Ours** | **87.3** | 68.5 | **78.6** | **66.0** | 62.4 | **78.2** | **53.1** |

**Comparison with baselines** table 2 presents the accuracy on TVQA+, MMSI, and OnlineMMSI. On TVQA+, our method improves Video Multiple-Choice QA accuracy by 2.1% over Qwen2.5-VL, achieving a new state-of-the-art result. On MMSI and OnlineMMSI, our approach yields gains of 4.0%, 2.5%, and 5.3% across three social tasks, demonstrating that our method significantly

enhances MLLMs' ability to understand social interaction. We observe that the improvements on MMSI are higher than on TVQA+. This is because MMSI videos involve more participants, highlighting the advantage of our approach in handling multi-speaker alignment under more complex scenarios. In addition, TVQA+ videos are drawn from scripted TV shows, where speaker characters are fixed and the model may learn name-token associations during finetuning. Compared to baselines that rely on injecting box coordinates, speaker names, or color cues into the text input to associate modalities, our method requires no such auxiliary prompts. While their improvements are often unstable across tasks (boosting performance on some tasks while degrading others), our method modifies attention distributions in a natural and direct manner, achieving stable and generalizable cross-modal alignment for social interaction tasks.

On the other hand, our method does not surpass the current state-of-the-art on the speaking target identification task, likely because this task requires more balancing attention between both the current speaker and the his speaking target. However, we still achieves the second-best accuracy with competitive performance, and on pronoun coreference resolution and mentioned speaker prediction, our approach significantly outperforms prior methods on MMSI and OnlineMMSI.

**Visualizations** We present visualizations of Qwen2.5-VL's cross-attention maps before and after applying our social-aware bias in fig. 4. As shown in example (a), when asked about the behavior of the character Penny, Qwen2.5-VL incorrectly predicted "raise hand", which is actually the action of another character, Beverley. The attention map reveals that a considerable portion of Penny's attention was misaligned to Beverley's region. After adding our bias, the attention naturally concentrates on Penny, leading to the correct answer "tap the bar".

In the case (b), the question concerns the emotion of Sheldon when switching beds (third image, corresponding to Sheldon's second utterance). We visualize the attention maps of the second "Sheldon" token across frames. Without our bias, Qwen2.5-VL assigns attention uniformly across Sheldon's visual tokens over all frames. By adding our bias, the model clearly emphasizes the third frame over the first, achieving more accurate spatial–temporal–speaker alignment between text and video, and producing the correct answer. Similarly, in two examples (c)(d) from MMSI, our bias enables precise modeling of current speaker in videos, further enhancing the understanding of social interactions.

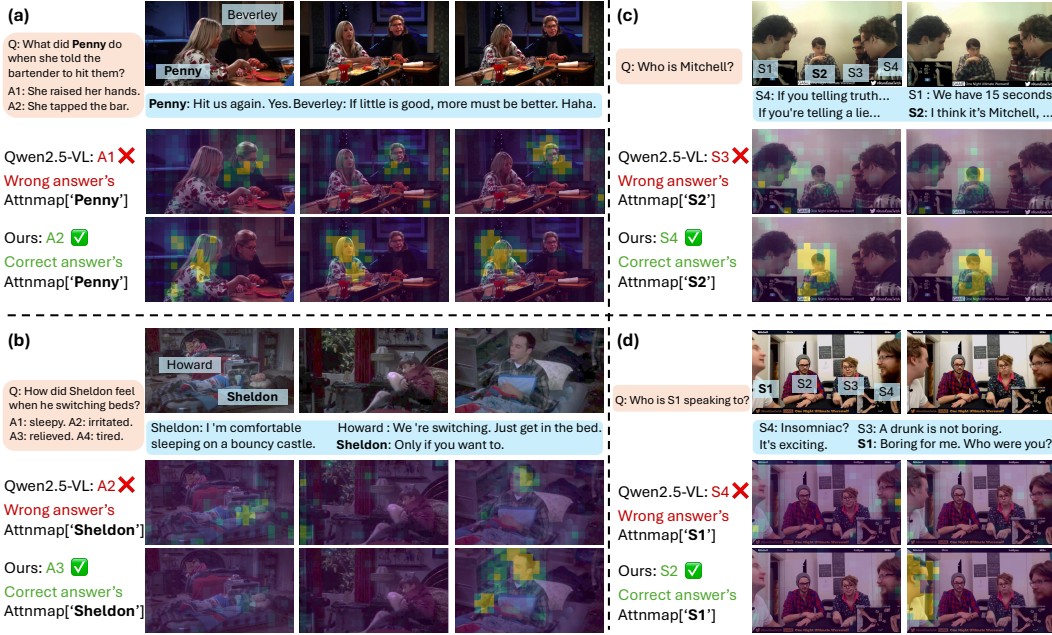

Figure 4: Attention maps in Qwen2.5-VL layer 16 before and after adding social-aware bias. Our bias enables more accurate spatial–temporal–speaker alignment. Video resolution is 640x360.

## 5.4 ABLATIONS

To examine the effectiveness of different components of our method, we conduct ablations on active head selection and social-aware bias.

**Transformer Layers** We investigate the effect of applying bias at different layers of the transformer, including all layers (0–27), as well as subsets of early, middle, and late layers. As shown in table 3, the best performance is achieved when the bias is applied to middle layers (10–19), followed by all layers. This finding suggests that middle layers may play a more crucial role in cross-modal feature fusion. This observation is consistent with prior studies (Zhang et al., 2025a; Liang et al., 2024b; Chen et al., 2025; Bi et al., 2025), as well as with our visualization analysis conducted on layer 16.

**Active Head Threshold** We vary the cross-attention strength threshold $\lambda$ and report the results with the ratio of active heads in table 4. Note that we only apply the bias to middle layers, thus the maximum ratio is 35.7%. We find that the best performance is achieved at a small threshold of $5e-5$. Compared to the original Qwen2.5-VL, even activating only 9% of heads yields an average improvement of about 3% across tasks, while activating 25% achieves a 4% gain. This demonstrates the importance of our bias in facilitating multi-speaker multimodal understanding. In contrast, activating all heads leads to a drop in performance, likely because some heads are responsible for attending positional encoding or text modality, while adding bias on them disrupts their stability.

Table 3: Effect of transformer layers.

| Layers | TVQA+ | MMSI | | |
|---|---|---|---|---|
| | *VideoQA* | *T* | *P* | *M* |
| 0-27 | 85.6 | 66.9 | **79.1** | 63.2 |
| 0-9 | 86.0 | 66.9 | 76.7 | 64.4 |
| 10-19 | **87.3** | **68.5** | 78.6 | **66.0** |
| 20-27 | 86.2 | 66.4 | 78.4 | 64.4 |

Table 4: Effect of the number of active heads.

| $\lambda$ | Active heads(%) | TVQA+ | MMSI | | |
|---|---|---|---|---|---|
| | | *VideoQA* | *T* | *P* | *M* |
| 0 | 35.7 | 85.6 | 65.4 | 77.8 | 61.2 |
| 5e-5 | 24.6 | **87.3** | **68.5** | 78.6 | **66.0** |
| 2e-4 | 15.8 | 86.8 | 67.9 | 78.2 | 66.0 |
| 8e-4 | 9.0 | 86.5 | 68.3 | **78.8** | 64.9 |
| inf | 0.0 | 85.1 | 63.3 | 77.2 | 58.3 |

**Bias Strength** We evaluate different strategies for setting the bias strength, with results shown in table 5. Compared to the fixed-value strategy, our adaptive $W_b$ in eq. (4) consistently achieves better performance. A fixed large bias forces the model to over-focus on the guided regions while ignoring global visual information, which in turn leads to a performance drop. This indicates that our adaptive social-aware biasing mechanism is highly natural: it enhances attention toward the current speaker's region without disrupting the model's inherent attention patterns, thereby improving cross-modal alignment and yielding stronger performance across social interaction tasks.

Table 5: Effect of bias strength.

| Bias | TVQA+ | MMSI | | |
|---|---|---|---|---|
| | *VideoQA* | *T* | *P* | *M* |
| *fixed* | | | | |
| 0 | 85.1 | 63.3 | 77.2 | 58.3 |
| 10 | 86.4 | 65.7 | 75.3 | 63.5 |
| 100 | 84.4 | 64.2 | 72.5 | 53.8 |
| *adaptive* | | | | |
| $0.5 \cdot max$ | 86.8 | 66.8 | 77.2 | 63.7 |
| $1 \cdot max$ | **87.3** | **68.5** | **78.6** | **66.0** |
| $2 \cdot max$ | 86.0 | 66.7 | 77.4 | 64.1 |

## 6 CONCLUSION

This paper presents a method to help multimodal large language models better understand multimodal multi-speaker social interactions. Building on a systematic analysis of cross-modal attention, the proposed method strengthens the alignment between visual and textual tokens belonging to the same speaker. Experiments across multiple datasets and tasks validate its effectiveness in improving multi-speaker reasoning. Future research directions include further investigating the role of attention heads in cross-modal alignment, exploring ways to leverage inherent grounding abilities of MLLMs to guide alignment without relying on bounding box annotations, thereby reducing annotation costs and enhancing efficiency for social AI.

**LLM Usage** In this work, a large language model (ChatGPT) was employed solely for language polishing and writing refinement. Its role was limited to improving clarity and readability of the manuscript. LLM was **not** involved in the design of the methodology, data processing, or analysis.

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

# A APPENDIX

## A.1 TASKS INSTRUCTIONS

We evaluate our method on four types of social interaction tasks. Each task takes as input the video frames, transcripts, and speaker bounding boxes. We describe the tasks and prompts used for MLLMs below:

- **Video Question Answering.** This task requires answering questions grounded in multi-party dialogue videos. In TVQA+ (Lei et al., 2020), each question is accompanied by five candidate choices. Instruction prompt: `<video>\nWatch this video of speakers social interaction, read their conversation, question and choose the correct answer. {Conversation}. Q: How does Sheldon feel? a0: tired, ..., a4: angry.`
- **Speaking Target Identification.** This task identifies the addressee (speaking target) of the current speaker in the dialogue. Instruction prompt: `<video>\nWatch this video of {N} speakers social interaction, be aware of their non-verbal behaviours. Read the conversation. {Conversation}. Predict the speaking target (speaking to whom) of this sentence: Speaker 0: Who were you?`
- **Pronoun Coreference Resolution.** This task aims to resolve pronouns in the dialogue transcripts to their corresponding speakers. Instruction prompt: `<video>\nWatch this video of {N} speakers social interaction, be aware of their non-verbal behaviours. Read the conversation. {Conversation}. Predict which speaker should be the 'he' in this sentence: Speaker 1: Did he not say that?`
- **Mentioned Player Prediction.** This task requires linking a dialogue mentioned name to the correct participant appearing in the video. Instruction prompt: `<video>\nWatch this video of {N} speakers social interaction, be aware of their non-verbal behaviours. Read the conversation. {Conversation}. Predict which speaker should be the 'Mitchell' in this sentence: Speaker 3: I think it's Mitchell.`

## A.2 BASELINE METHODS

We compare our method against several baseline approaches in table 2, described as follows:

- **Random.** For the VQA task, the model randomly selects one answer from five candidates. For MMSI tasks, it randomly selects one speaker among $N$ candidates.
- **ST-VLM-7B (Ko et al., 2025) and TLNet (Liang et al., 2024a)** are highly competitive models on TVQA+. However, since our setting requires at least one speaker with bounding box annotations, our training and test sets differ from theirs. Despite this difference, our method substantially outperforms these baselines.
- **MMSI (Lee et al., 2024a)** employs a transformer to align and fuse text features from language models with visual interaction features derived from bounding boxes and keypoints. Classification tasks are then performed via a masked modeling objective to solve three social interaction tasks. We report their best-performing result, i.e., the RoBERTa-based baseline, for comparison.
- **OnlineMMSI (Li et al., 2025a)** leverages bounding boxes and keypoints as visual prompts to align multiple speakers for MLLMs. Since the implementation was not publicly released, we re-implemented it based on Qwen2.5-VL, using only bounding box annotations for a fair comparison. Notably, our reproduced results are even higher than original reported numbers, which may be due to preprocessing differences, but this does not affect the fairness of comparison.
- **Qwen2.5-Text (Bai et al., 2025)** is a text-only baseline where Qwen2.5-VL is given only the dialogue transcripts without any video input.
- **Qwen2.5-VL (Bai et al., 2025)** is a strong MLLM baseline. We use the instruction prompt described in appendix A.1, but additionally append bounding box coordinates as text prompts, e.g., `Speaker 1 [100,100,300,400]`.

