# OpenReview forum: "Multimodal Social Interaction with Multi-speaker Attention Alignment"
_ICLR.cc/2026/Conference — ICLR 2026 Conference Withdrawn Submission_

### Official Review · Reviewer_KD4L · 2025-10-22

**Soundness:** 3
**Presentation:** 4
**Contribution:** 2
**Rating:** 4
**Confidence:** 3

**Summary:**

This paper investigates the unstable performance of Multimodal Large Language Models (MLLMs) in understanding social interactions in videos. The authors observe a significant misalignment between the attention focus of MLLMs and the actual positions of speakers in the cross-modal attention patterns of text and vision tokens within MLLMs. To address this issue, they propose Dynamic Cross-Modal Head Selection and Adaptive Social-Aware Attention Bias to correct the attention patterns in MLLMs. The effectiveness of the proposed method, manifested in certain performance improvements, has been verified on multiple relevant benchmarks including MMSI and OmniMMSI.

**Strengths:**

1. This paper identifies the shortcomings of existing Multimodal Large Language Models (MLLMs) in understanding social interactions in videos, proposes a direct solution, effectively achieves its research motivation, and obtains convincing results.

2. The ablation experiments and visualization results of this paper are comprehensive. The writing is well-structured and easy to follow.

**Weaknesses:**

1. The method in this paper is limited by pre-provided speaker bounding boxes, which I believe greatly reduces the method’s generality and practicality.

2. Intuitively, I think the model can be trained to focus on the correct speaker positions through better training data, rather than relying on training-free corrections (though I acknowledge the latter is effective in many cases). In other words, as the model’s capability further improves, I am concerned about whether the method proposed in this paper will still be effective. I look forward to seeing results on larger-scale models (e.g., 72B parameters) and more powerful models (such as Qwen-3VL).

**Questions:**

1. Is it feasible to pre-provide speaker bounding boxes in real-world application scenarios? I think this seems like a supervision signal that is difficult to obtain.

2. Can we first use human detection to obtain weak annotations of human positions, and then use the method in this paper to enhance the attention between speakers and humans (without requiring precise correspondence)? I believe adding such a solution can greatly enhance practicality.

---

### Official Review · Reviewer_MtN5 · 2025-10-31

**Soundness:** 2
**Presentation:** 2
**Contribution:** 2
**Rating:** 2
**Confidence:** 5

**Summary:**

This paper tackles a familiar failure in MLLM watching multi-speaker video. When several people talk, the model’s text tokens do not consistently lock onto the right speaker’s visual region, so grounding drifts and answers wobble. The authors propose two steps. They first pick attention heads that actually seem to carry cross-modal grounding. They then add a social-aware bias inside those heads so that words attributed to a given speaker pull attention toward that speaker’s visual tokens at the right time. The change happens at the score level, not by adding new weights. On TVQA+, MMSI, and OnlineMMSI they report steady gains and show attention maps that look more speaker-focused. The approach assumes access to speaker boxes and token-level speaker labels during training and evaluation.

**Strengths:**

1. The method is precise and simple to implement. The head selection rule and the bias formulation are clearly described and easy to reproduce.
2. The paper gives a sensible diagnostic for the problem and backs it with visuals and measurements that show scatter in multi-speaker scenes and tighter focus after the edit.
3. The reported gains span several social tasks and three datasets with a strong modern backbone, which suggests the tweak is practically useful in the supervised setting the paper targets.

**Weaknesses:**

1. It needs accurate speaker boxes and speaker labels for text tokens, and there is no test of how it holds up when those are noisy or missing.
2. The claim of being parameter-free hides real dependencies. For example.  the behavior is driven by privileged annotations and threshold choices, and there is no sensitivity study beyond the basic heuristic.
3. The paper calls the overhead minimal but gives no timing or memory numbers for head screening and biasing across realistic clips and batches. (Compute FLOPS, MACs, latency....)
4. The main tables lack confidence intervals or significance tests, so it is hard to judge reliability across seeds and videos.
5. All results come from curated datasets with clean boxes and aligned transcripts, leaving open how the approach handles off-screen speech, occlusion, or diarization errors in the wild.

**Questions:**

1. How robust is the method when boxes are noisy, drift, or are missing for some frames? Please simulate realistic annotation errors and dropouts and report how accuracy degrades
2. What is the true cost of the intervention? Please report end-to-end latency and peak memory for typical clip lengths and batch sizes, with and without the bias step.
3. How sensitive are results to the head-selection threshold and to the number of selected heads? A sweep plotting accuracy versus selected-head count and a stability check across seeds would help.
4. Does the bias hurt cases where the answer depends on non-speaking entities or background context rather than the current speaker? Include targeted stress tests.
6. Can you explain the need for token-level speaker labels? If not, how often are tokens mislabeled, and how do those errors propagate. If yes, show an automatic linking variant and measure the gap.
7. Do the gains persist on another strong video-capable MLLM under the same protocol, so we can rule out backbone-specific effect?

---

### Official Review · Reviewer_xkZM · 2025-11-01

**Soundness:** 3
**Presentation:** 3
**Contribution:** 3
**Rating:** 4
**Confidence:** 3

**Summary:**

This paper identifies and quantifies cross-modal attention misalignment in MLLMs as a key bottleneck for multi-party social interaction understanding. The authors propose a novel, parameter-free attention alignment method that dynamically strengthens the association between speakers' visual appearances and their textual utterances. Extensive experiments show that the method improves performance across diverse multimodal social interaction tasks.

**Strengths:**

+ The paper is well-constructed
+ This paper provides a clear analysis of the shortcomings of MLLM for multi-party social interaction understanding.

**Weaknesses:**

- The impact of cross-modal head selection. Deactivating some heads may result in the loss of some useful information, thus impacting performance improvements on certain tasks. Further analysis is recommended.
- Hyperparameter selection. The proposed method introduces several hyperparameters, such as the fixed threshold lambda. Although the authors present ablation studies for these hyperparameters, they do not explain how to choose them in practice. Whether they are selected via validation set search?
- Model scalability. Can the proposed method be applied to different Video MLLMs? How does it perform on better baselines, such as Qwen3-VL or models with a larger number of parameters? Are the hyperparameters robust?
- Fairness of the comparison. Sections 5.1 and 5.2 mention the need to use LoRa for training, but don't explain whether the comparison method has been trained. More detailed explanations are recommended.
- Attention-based MLLM. Various methods, including training-free and training-based approaches, have been proposed to modify attention weights in attention-based MLLMs. However, there is a lack of comprehensive discussion and comparison of these methods. Furthermore, these methods share significant similarities with the approach proposed in this study.

**Questions:**

see weakness

---

### Official Review · Reviewer_9Zif · 2025-11-01

**Soundness:** 3
**Presentation:** 2
**Contribution:** 2
**Rating:** 6
**Confidence:** 2

**Summary:**

This paper identifies a key limitation of MLLMs in multi-speaker video understanding: weak cross-modal alignment between speakers' visual regions and their utterances. To address this, the authors propose a training-free attention alignment method that dynamically selects relevant attention heads and applies a social-aware bias to reinforce speaker-specific visual-text correspondences. Evaluated on three datasets across four social reasoning tasks, the approach yields consistent improvements and achieves state-of-the-art results, demonstrating its effectiveness in enhancing multi-modal social interaction understanding.

**Strengths:**

* The method proposes dynamic cross-modal head selection and an adaptive social-aware attention bias. It does not disrupt the original attention pattern, requires no additional training, and accurately enhances image-text (speech-translated text) alignment in multi-speaker social interaction scenarios, which is confirmed in Table 1.
* This paper conducts insightful ablation studies to analyze the impacts of bias strength, transformer layer selection, and the maximum number of active heads.

**Weaknesses:**

* The method’s reliance on ground-truth speaker bounding boxes limits its generalization to realistic settings where boxes may be missing, overlapping, or misaligned, robustness under such conditions remains unevaluated.
* Only Qwen2.5-VL is employed as the backbone, and validation has not been conducted on other Multimodal Large Language Model (MLLM) architectures such as the GLM series. As a result, the generalizability of the method and the scope of conclusion extrapolation remain limited.
* The method is training-free and effective; however, the lack of a publicly available code repository undermines its reproducibility.

**Questions:**

* How does this method handle fully open-world or unsupervised speaker recognition scenarios, i.e., situations where speaker bboxes or names are not pre-annotated?
* Could the bias mechanism degrade performance when speakers are spatially adjacent or their bboxes overlap? Providing failure cases or attention maps for such ambiguous scenes would clarify this risk.

---

### Note · Authors · 2025-11-12

I have read and agree with the venue's withdrawal policy on behalf of myself and my co-authors.